[Supplementary Material]

# A    When Do Bigger Models Help More?

Figure A.1 shows relative improvement by increasing the model size under different amount of labeled examples. Both supervised learning and semi-supervised learning (i.e. SimCLRv2) seem to benefit from having bigger models. The benefits are larger when (1) regularization techniques (such as augmentation, label smoothing) are used, or (2) the model is pretrained using unlabeled examples. It is also worth noting that these results may reflect a "ceiling effect": as the performance gets closer to the ceiling, the improvement becomes smaller.

Figure A.1:   Relative improvement (top-1) when model size is increased.  (a) supervised learning without extra regularization, (b), Supervised learning with auto-augmentation [35] and label smoothing [71] are applied for 1%/10% label fractions, (c) semi-supervised learning by fine-tuning SimCLRv2.

# B    Parameter Efficiency Also Matters

Figure B.1 shows the top-1 accuracy of fine-tuned SimCLRv2 models of different sizes. It shows that (1) bigger models are better, but (2) with SK [28], better performance can be achieved with the same parameter count. It is worth to note that, in this work, we do not leverage group convolution for SK [28] and we use only $3 \times 3$ kernels. We expect further improvement in terms of parameter efficiency if group convolution is utilized.

Figure B.1:   Top-1 accuracy of fined-tuned SimCLRv2 models of different sizes on three label fractions. ResNets with depth in $\{50, 101, 152\}$, width in $\{1\times, 2\times\}$ are included here. Parameter efficiency also plays an important role. For fine-tuning on 1% of labels, SK is much more efficient.

# C    The Correlation Between Linear Evaluation and Fine-tuning

Most existing work [17, 19, 18, 42, 20, 1] on self-supervised learning leverages linear evaluation as a main metric for evaluating representation quality, and it is not clear how it correlates with semi-supervised learning through fine-tuning. Here we further study the correlation of fine-tuning and linear evaluation (the linear classifier is trained on the ResNet output instead of some middle layer of projection head). Figure C.1 shows the correlation under two different fine-tuning strategies:

fine-tuning from the input of projection head, or fine-tuning from a middle layer of projection head. We observe that overall there is a linear correlation. When fine-tuned from a middle layer of the projection head, we observe a even stronger linear correlation. Additionally, we notice the slope of correlation becomes smaller as number of labeled images for fine-tuning increases .

Figure C.1:   The effects of projection head for the correlation between fine-tuning and linear evaluation. When allowing fine-tuning from the middle of the projection head, the linear correlation becomes stronger. Furthermore, as label fraction increases, the slope is decreasing. The points here are from the variants of ResNets with depth in $\{50, 101, 152\}$, width in $\{1\times, 2\times\}$, and with/without SK.

## D   The Impact of Memory

Figure D.1 shows the top-1 comparisons for SimCLRv2 models trained with or without memory (MoCo) [20]. Memory provides modest advantages in terms of linear evaluation and fine-tuning with 1% of the labels; the improvement is around 1%. We believe the reason that memory only provides marginal improvement is that we already use a big batch size (i.e. 4096).

Figure D.1:   Top-1 results of ResNet-50, ResNet-101, and ResNet-152 trained with or without memory.

## E   The Impact of Projection Head Under Different Model Sizes

To understand the effects of projection head settings across model sizes, Figure E.1 shows effects of fine-tuning from different layers of 2- and 3-layer projection heads. These results confirm that with only a few labeled examples, pretraining with a deeper projection head and fine-tuning from a middle layer can improve the semi-supervised learning performance. The improvement is larger with a smaller model size.

(a) w/o SK (label fraction: 1%)    (b) w/o SK (label fraction: 10%)    (c) w/o SK (label fraction: 100%)

(d) w/ SK (label fraction: 1%)    (e) w/ SK (label fraction: 10%)    (f) w/ SK (label fraction: 100%)

Figure E.1: Top-1 fine-tuning performance under different projection head settings (number of layers included for fine-tuning) and model sizes. With fewer labeled examples, fine-tuning from the first layer of a 3-layer projection head is better, especially when the model is small. Points reflect ResNets with depths of $\{50, 101, 152\}$ and width multipliers of $\{1\times, 2\times\}$. Networks in the first row are without SK, and in the second row are with SK.

Figure E.2 shows fine-tuning performance for different projection head settings of a ResNet-50 pretrained using SimCLRv2. Figure 5 in the main text is an aggregation of results from this Figure.

Figure E.2: Top-1 accuracy of ResNet-50 with different projection head settings. Deeper projection head help more, when allowing to fine-tune from a middle layer of the projection head.

# F    Further Distillation Ablations

Figure F.1 shows the impact of distillation weight ($\alpha$) in Eq. 3, and temperature used for distillation. We see distillation without actual labels (i.e. distillation weight is 1.0) works on par with distillation

with actual labels. Furthermore, temperature of 0.1 and 1.0 work similarly, but 2.0 is significantly worse. For our distillation experiments in this work, we by default use a temperature of 0.1 when the teacher is a fine-tuned model, otherwise 1.0.

(a) label fraction: 1%  (b) Label fraction: 10%

Figure F.1: Top-1 accuracy with different distillation weight ($\alpha$), and temperature ($\tau$).

We further study the distillation performance with teachers that are fine-tuned using different projection head settings. More specifically, we pretrain two ResNet-50 (2×+SK) models, with two or three layers of projection head, and fine-tune from a middle layer. This gives us five different teachers, corresponding to different projection head settings. Unsurprisingly, as shown in Figure F.2, distillation performance is strongly correlated with the top-1 accuracy of fine-tuned teacher. This suggests that a better fine-tuned model (measured by its top-1 accuracy), regardless their projection head settings, is a better teacher for transferring task specific knowledge to the student using unlabeled data.

(a) Label fraction: 1%  (b) Label fraction: 10%

Figure F.2: The strong correlation between teacher's task performance and student's task performance.

# G   CIFAR-10

We perform main experiments on ImageNet since it is a large-scale and well studied dataset. Here we conduct similar experiments on small-scaled CIFAR-10 dataset to test our findings in ImageNet. More specifically, we pretrain ResNets on CIFAR-10 without labels following [1] [5]. The ResNet variants we trained are of 6 depths, namely 18, 34, 50, 101, 152 and 200. To keep experiments tractable, by default we use Selective Kernel, and a width multiplier of 1×. After the models are pretrained, we then fine-tune them (using simple augmentations of random crop and horizontal flipping) on different numbers of labeled examples (250, 4000, and total of 5000 labeled examples), following MixMatch's protocol and running on 5 seeds.

The fine-tuning performances are shown in the Figure G.1, and suggest similar trends to our results on ImageNet: big pretrained models can perform well, often better, with a few labeled examples. These

Figure G.1: Fine-tuning pre-trained ResNets on CIFAR-10.

results can be further improved with better augmentations during fine-tuning and an extra distillation step. The linear evaluation also improves with a bigger network. Our best result is 96.4% obtained from ResNet-101 (+SK) and ResNet-152 (+SK), but it is slightly worse (96.2%) with ResNet-200 (+SK).

# H   Extra Results

Table H.1 shows top-5 accuracy of the fine-tuned SimCLRv2 (under different model sizes) on ImageNet.

Table H.1: Top-5 accuracy of fine-tuning SimCLRv2 (on varied label fractions) or training a linear classifier on the ResNet output. The supervised baselines are trained from scratch using all labels in 90 epochs. The parameter count only include ResNet up to final average pooling layer. For fine-tuning results with 1% and 10% labeled examples, the models include additional non-linear projection layers, which incurs additional parameter count (4M for $1\times$ models, and 17M for $2\times$ models).

| Depth | Width | Use SK [28] | Param (M) | Fine-tuned on | | | Linear eval | Supervised |
|---|---|---|---|---|---|---|---|---|
| | | | | 1% | 10% | 100% | | |
| 50 | $1\times$ | False | **24** | **82.5** | **89.2** | **93.3** | **90.4** | **93.3** |
| | | True | 35 | 86.7 | 91.4 | 94.6 | 92.3 | 94.2 |
| | $2\times$ | False | 94 | 87.4 | 91.9 | 94.8 | 92.7 | 93.9 |
| | | True | 140 | 90.2 | 93.7 | 95.9 | 93.9 | 94.5 |
| 101 | $1\times$ | False | 43 | 85.2 | 90.9 | 94.3 | 91.7 | 93.9 |
| | | True | 65 | 89.2 | 93.0 | 95.4 | 93.1 | 94.8 |
| | $2\times$ | False | 170 | 88.9 | 93.2 | 95.6 | 93.4 | 94.4 |
| | | True | 257 | 91.6 | 94.5 | 96.4 | 94.5 | 95.0 |
| 152 | $1\times$ | False | 58 | 86.6 | 91.8 | 94.9 | 92.4 | 94.2 |
| | | True | 89 | 90.0 | 93.7 | 95.9 | 93.6 | 95.0 |
| | $2\times$ | False | 233 | 89.4 | 93.5 | 95.8 | 93.6 | 94.5 |
| | | True | 354 | 92.1 | 94.7 | 96.5 | 94.7 | 95.0 |
| 152 | $3\times$ | True | **795** | **92.3** | **95.0** | **96.6** | **94.9** | **95.1** |

## Footnotes

[5]For CIFAR-10, we do not find it beneficial to have a 3-layer projection head, nor using a EMA network. However, we expand augmentations in [1] with a broader set of color operators and Sobel filtering for pretraining. We pretrain it for 800 epochs, with a batch size of 512 and temperature of 0.2.