[Reviews · NeurIPS 2020]

Review 1

Summary and Contributions: This empirical work proposes a semi-supervised framework that consists of three steps: self-supervised pretraining, fine-tuning, and distillation. The framework achieves SOTA on Imagenet for semi-supervised learning.

Strengths: The work pushes current SOTA on ImageNet benchmark for semi-supervised learning. The main contribution is an unlabelled distillation over self-supervised learning models.

Weaknesses: The framework is mostly incremental. The main changes are increased capacity and distillation of the neural network. Specifically, the authors utilize 152-layer ResNet with a wider layer and modification with SK kernels. While the SOTA results on Imagenet was introduced using EfficientNets by Xie et al., CVPR 2020, authors could utilize better deep architectures. Improving the projection head is mostly hyperparameter optimization of the SimCLR framework. Because the flexible performance is seen in the ablation study for example in Hjelm et al. ICLR 2019 or can be further explained by an idea of Information bottleneck by Tishby et al. The performance depends on the layer from which features were taken. MoCo modification for comparison is again a hyperparameter optimization contribution. Overall, there is no significant contribution to unsupervised pre-training.

Correctness: The main claim about the unreasonable effectiveness of big models is supported only with empirical results on ImageNet. It also has not been discussed with current literature. Specifically, authors may explore the direction of The Lottery Ticket Hypothesis by Jonathan Frankle (ICLR 2019).

Clarity: The manuscript is well-written.

Relation to Prior Work: The authors distinctly discussed literature related to unsupervised learning and distillation. However, there is a lack of related work about learning with hugely over-parametrized deep neural networks, which can lead to more discussion about the raised question.

Reproducibility: Yes

Additional Feedback: Thank you for the effort in rebuttal. I decided to raise the score. The authors empirically showed that distillation from big models should be a valuable and important step for the downstream task with self-supervised methods.


Review 2

Summary and Contributions: The paper shows that unsupervisedly learned representation encoder, with a small fraction of labels, can achieve very good classification performance on ImageNet. --- Update: After reading the authors' rebuttal and other reviews, I decide to keep my original score.

Strengths: The result is quite exciting, showing the benefit of a good representation in terms of label efficiency. This could allow more work to explore using representation encoders not just for transferring to downstream tasks, but for reducing labelling effort too.

Weaknesses: I have several questions regarding to the method: 1. If I understand correctly, the combination of using a larger MLP projection head, and finetuning from the middle layer of the MLP head, is the same as just adding one more FC layer to the base network. Is this true? If so, I suggest the authors revise relevant parts of lines 103-130 because the current wording is both more confusing, and seems to imply a bigger contribution than simply adding a layer. Moreover, in this case, the contrastive learning part should be more properly described as using a combination of MoCo and SimCLR, rather than a variant of only the latter, which contributes MLP head and augmentations. 2. What exactly is named SimCLRv2? The contrastive learning part, or the entire method (including finetuning & distillation)? The paper refers to both with SimCLRv2 in various places. If it is the latter, I suggest using a different name because it is solving a different task.

Correctness: The claims and the evaluation methods seem correct.

Clarity: The paper is generally well written other than the parts that let to my questions described above.

Relation to Prior Work: Related work is properly discussed.

Reproducibility: Yes

Additional Feedback:


Review 3

Summary and Contributions: This paper proposes a simple semi-supervised paradigm: 1) self-supervised pretrain, 2) fine-tune with labeled data, 3) distillation via unlabeled data. The simple method achieves very good performance outperforming prior arts with large margins. Besides the necessary of distillation for semi-supervised learning, authors also demonstrate how to use big model and projection head to further help semi-supervised learning performance, leading to SimCLR v2.

Strengths: + The paper is well written and easy to follow. + The proposed method is simple yet efficient. Experimental results are quite strong. + Good ablation study.

Weaknesses: - Most major parts in this work, such as distillation, fine-tuning are proposed in previous works. Although the authors improve SimCLR and proposed SimCLR V2, the novelties in individual parts are somehow limited. However, I think the simple semi-supervised framework is still valuable for industry and future works. It explicitly points out a previously ignored paradigm in semi-supervised visual learning where regularization based methods dominates. I think it will inspire several future works following the paradigm. The finding of big model can help semi-supervised learning is also very valuable for practical usage. - My major concern is that this paper only conducts experiments on ImageNet, while recent semi-supervised papers are also benchmarked in several datasets (CIFAR10/100, SVHN, etc). And an important conclusion in this paper is that "big model is good for semi-supervised learning". In small dataset, it is well known big models are hard to train. Thus a natural question comes: is this conclusion still valid for small datasets? How big the dataset should be to effectively use your proposed method (either semi-supervised or unsupervised method)? - Some minor things: 1) The gap between Fig. 1 and Fig. 2 can be enlarged to avoid confusion. 2) For Table 3, although I know it's fine-tuning the self-supervised model, I suggest authors to add descriptions about "Methods using self-supervised representation learning only" to avoid confusion. - Lack of references. Scaling and Benchmarking Self-Supervised Visual Representation Learning, ICCV 2019 Learning deep representations by mutual information estimation and maximization, ICLR 2019 Confidence Regularized Self-Training, ICCV 2019 - My another concern is that more insights about distillation for semi-supervised learning should be provided. 1) The ablation in Table 2 of 1% labels shows that distillation can provide 50%+ performance gain which is quite surprising. The student model is much better than the teacher model. Are there any insights and explanations why this happen? Is augmentation important in this case? 2) Moreover, a smaller model has worse semi-supervised performance than self-distillated bigger model. This is also surprising. Are there any insights? 3) In addition, if self-distillation loss is used, the gradient of the self-distillation loss w.r.t. logits should be P_T - P_S, why the distillation can bring so much improvement compared to teacher network? ------------------------------------ Updates: The rebuttal partially addresses my concerns, especially on different datasets. And after reading other reviewers' comments, I think this paper shows strong performances on semi-supervised benchmarks, using an ignored simple method. Thus I would like to raise my score.

Correctness: Yes

Clarity: A well written paper.

Relation to Prior Work: Relations to prior works are clearly discussed.

Reproducibility: Yes

Additional Feedback: 1. I think if the paper wants to claim "bigger models are good for semi-supervised learning", the proposed methods should be evaluated on several benchmarks, including small datasets. If the proposed claim is not true on small datasets, such a claim can be misleading to the community and should be modified to 'large-scale semi-supervised learning'. But if such claim is true for small datasets, my guess is that a relative bigger model can still help semi-supervised learning on small dataset. But the authors should show the exploration on these datasets. This is also a practical concern since in some settings (medical) where the entire dataset is small. 2. It's interesting to see the semi-supervised trained models' transfer learning performances on downstream tasks, using fine-tuning. The proposed semi-supervised models may be very good pretrained models for downstream tasks. Although I appreciate the extensive experiments on ImageNet, but my initial rating is "Marginal below acceptance".


Review 4

Summary and Contributions: This paper proposes SimCLRv2, an extension of SimCLR[1] to the semi-supervised learning problem. SimCLRv2 comprises three steps: (1) self-supervised pretraining; (2) supervised finetuning; (3) model distillation. The key findings of the paper include (i) bigger models are more data-efficient (ii) the big pretrained model can be distilled into a smaller model with little loss in performance. SimCLRv2 outperforms previous SOTA on semi-supervised learning by a large margin.

Strengths: (1) The performance of SimCLRv2 is impressive, e.g. when using 1% labeled data, SimCLRv2 distilled resnet50 model obtains 73.9 top1 accuracy, which is close to the fully supervised resnet50 model (100% labeled data, 76.5 top1 accuracy) (2) SimCLR[1] demonstrates good representation can be learned without using labels. SimCLRv2 pushes one step further, and shows that, by using a much bigger model, much better representation can be learned without using labels (still requires a lot of unlabeled data and computational resources though). The finding is intriguing. (3) The paradigm of pretraining + finetuning a big model, then distilling it into a smaller model is new for semi-supervised learning (especially the distillation part), and achieves surprisingly good performance. (4) Extensive experiments have been run to justify the claims.

Weaknesses: All experiments are conducted on ImageNet, which is a curated dataset, with centered objects, well-balanced class/image distribution. In a realistic semi-supervised setting, as discussed in [43], the labeled/unlabeled data may be less curated, exhibit imbalanced/out-of-distribution class/image samples, etc. Such uncurated data is a natural testbed for the robustness of the model. It would be better if more discussion on realistic evaluation of the model is included.

Correctness: The proposed method is technically sound. The claims are well justified.

Clarity: The paper is well written.

Relation to Prior Work: The discussion on the related work is comprehensive.

Reproducibility: Yes

Additional Feedback: ------------------------------------ Post-rebuttal While the reviewers still have a concern about novelty, I'd like to keep my rating, given the strong empirical results.

[Author Response · NeurIPS 2020]

**Author Response for "The Unreasonable Effectiveness of Big Models for Semi-Supervised Learning"**

We thank the reviewers for feedback, as well as efforts in reviewing. We respond to each comment below.

**R1**: *"The framework is mostly incremental . . . Overall, there is no significant contribution to unsupervised pre-training."*
**R3**: *"Most major parts in this work . . . are proposed in previous works...the novelties in individual parts are somehow*
*limited."* The fact that our main contribution is a detailed procedure, rather than a theorem, architecture, or other artifact,
does not make the contribution less novel. Compared to previous work, our procedure achieves a $10\times$ improvement
in label efficiency on ImageNet. That we are able to obtain such a large improvement on a well-studied benchmark
indicates that our procedure was not previously known to the research community. Beyond the procedure itself, we
provide an extensive empirical evaluation of the importance of model size, and ablations for our specific design choices.

We believe our contributions are significant. Our results set a new, much higher bar for future work, and our procedure
is likely to have major practical applications. Indeed, R3 recognizes that "the simple semi-supervised framework is still
valuable for industry. . . I think it will inspire several future works." The fact that our approach is simple and intuitive
should count in favor of it rather than against it.

**R1**: *"The main claim...is supported only with empirical results on*
*ImageNet."* **R3**: *"My major concern is that this paper only conducts*
*experiments on ImageNet... In small dataset, it is well known big*
*models are hard to train."* While we believe ImageNet is a much more
difficult and convincing dataset, we have conducted further experi-
ments on CIFAR-10 given the reviewers' concern. More specifically,
we pretrain ResNet on CIFAR-10 without labels. The ResNet variants
we trained are of 6 depths, namely 18, 34, 50, 101, 152 and 200. To
keep experiments tractable, by default we use Selective Kernel, and
a width multiplier of $1\times$. After the models are pretrained, we then
fine-tune them (using simple augmentations of random crop and hori-
zontal flipping) on different numbers of labeled examples (250, 4000,
and total of 5000 labeled examples), following MixMatch's protocol
and running on 5 seeds. The results are shown in the Figure 1, and
suggest similar trends to our results on ImageNet: big pretrained
models can perform well, often better, with a few labeled examples.

**Figure 1:** CIFAR-10 fine-tuning.

These results can be further improved with better augmentations during fine-tuning and an extra distillation step.

**R2**: *"If I understand correctly, the combination of using a larger MLP projection head, and finetuning from the middle*
*layer of the MLP head, is the same as just adding one more FC layer to the base network."* Fine-tuning from the first
layer of the MLP head is the same as adding an FC layer to the base network and removing an FC layer from the head.
The impact of this extra FC layer is contingent on the label fraction during fine-tuning (Figure 5b). We will clarify this.

**R2**: *"What exactly is named SimCLRv2? The contrastive learning part, or the entire method (including finetuning &*
*distillation)?"* We understand the confusion regarding our naming. The name "SimCLRv2" is intended for the whole
procedure (including fine-tuning and distillation with unlabeled data), and we will say this explicitly.

**R3**: *"The ablation in Table 2 of 1% labels shows that distillation can provide 50%+ performance gain which is quite*
*surprising. The student model is much better than the teacher model. Are there any insights and explanations why this*
*happen?"* The student model in Table 2 is not better than the teacher. The teacher (pretrained and fine-tuned) gets 70.6%
with 1% of the data and 77.0% with 10%, as shown in in Table 1. We will add the teacher's accuracy to Table 2. The
50%+ improvement is over pure supervised training on 1% of data (which is heavily overfitting). *"...if self-distillation*
*loss is used...why the distillation can bring so much improvement compared to teacher network?"* The improvement
from self-distillation is meaningful but modest: 74.9%→76.6% with 1% of labels and 80.1%→80.9% with 10% of
labels for our biggest architecture. It may result either from the use of weaker augmentation during distillation or from
regularization induced by distillation. We will add more discussion.

**R3**: *"It's interesting to see the semi-supervised trained models' transfer learning performances on downstream tasks,*
*using fine-tuning."* In this work, we focus primarily on the problem of semi-supervised learning. But we agree it is
interesting to look at the transfer performance of these semi-supervised models, and we will do so after the rebuttal
period. We will also address rest of the minor issues (e.g. additional references) brought up by the reviewer.

**R4**: *"All experiments are conducted on ImageNet, which is a curated dataset, with centered objects, well-balanced*
*class/image distribution."* We agree that ImageNet is a well curated dataset, and may not reflect all real world scenarios.
However, ImageNet is also a well-studied benchmark dataset, and in that sense, it is ideal for evaluating our algorithms
on. Nonetheless, we agree this is an important point, and will discuss this issue in the revision.

[Meta-Review · NeurIPS 2020]

All reviewers agree that this work pushes current SOTA on ImageNet benchmark for semi-supervised learning. Though the method is incremental, the overall framework shows very strong experimental results. Thus, all reviews tend to accept the paper. The concerns of R1 in the discussion part have been taken into consideration for final decision.